# New Parallel and Streaming Algorithms for Directed Densest Subgraph

**Slobodan Mitrović\***
University of California, Davis
smitrovic@ucdavis.edu

**Theodore Pan\***
University of California, Davis
thjpan@ucdavis.edu

**Mahdi Qaempanah\***
Sharif University of Technology
mahdi.ghaempanah111@student.sharif.edu

**Mohammad Amin Raeisi\***
Yale University
amin.raeisi@yale.edu \*

## Abstract

Finding dense subgraphs is a fundamental problem with applications to community detection, clustering, and data mining. Our work focuses on finding approximate densest subgraphs in *directed graphs* in computational models for processing massive data. We consider two such models: Massively Parallel Computation (MPC) and semi-streaming. We show how to find a $(2 + \varepsilon)$-approximation in $\tilde{O}(\sqrt{\log n})$ MPC rounds with sublinear memory per machine. This improves the state-of-the-art results by Bahmani et al. [BGM14, WAW 2014] and Mitrović & Pan [MP24, ICML 2024]. Moreover, we show how to find an $O(\log n)$-approximation in a *single pass* in semi-streaming. This is in stark contrast to prior work, which implies $\tilde{\Omega}(n^{1/6})$ approximation for a single pass; a better approximation is known only for randomized streams (Mitrović & Pan). This is the first *deterministic single-pass* semi-streaming algorithm for the densest subgraph problem, both for undirected and directed graphs. Our semi-streaming approach is also an insertion-only dynamic algorithm, attaining the first directed densest subgraph algorithm with $O(\log^2 n)$ worst-case update time while using sub-linear memory. We empirically evaluate our approaches in two ways. First, we illustrate that our single-pass semi-streaming algorithm performs much better than the theoretical guarantee. Specifically, its approximation on temporal datasets matches the $(2 + \varepsilon)$-approximation of an $O(\log n)$-pass algorithm by Bahmani et al. [BKV12, VLDB 2012]. Second, we demonstrate that our MPC algorithm requires fewer rounds than prior work.

## 1 Introduction

Computing dense subgraphs in directed graphs is a classical optimization task where we are interested in subgraphs with a large edge-to-vertex ratio. In particular, given a directed graph $G = (V, E)$, the densest subgraph (DS) problem asks to find two vertex subsets $S, T \subseteq V$ such that the ratio $|E(S, T)| / \sqrt{|S| \cdot |T|}$ is maximized, where $E(S, T)$ is the set of directed edges from $S$ to $T$. This problem has found a wide range of applications in graph mining, including the analysis of social networks [LRJA10, For10, CS10], bioinformatics [FNBB06, SHK+10], visualization [ZP12, ZT12], and finance [ZZY+17, FRM19, CT22, JZT+22].

With the constant increase in the size and prevalence of large datasets, it has become crucial to develop algorithms that solve fundamental optimization problems with limited constraints, such

---

\*Equal contribution.

39th Conference on Neural Information Processing Systems (NeurIPS 2025).

| Massively Parallel Computation model | | | | |
|---|---|---|---|---|
| | Approximation | Memory per machine | Round complexity | Reference |
| Undirected | $(1+\varepsilon)$ | $O(n^\delta)$ | $\tilde{O}(\sqrt{\log n})$ | [GLM19] |
| Directed | $(1+\varepsilon)$ | $O(n^\delta)$ | $O(\log n)$ | [BGM14] |
| | $(2+\varepsilon)$ | $O(n^\delta)$ | $\tilde{O}(\sqrt{\log n})$ | Our work |
| | $(2+\varepsilon)$ | $\tilde{O}(n)$ | $O(\sqrt{\log n})$ | [MP24] |
| Semi-streaming model | | | | |
| | Approximation | # of passes | Deterministic or randomized | Reference |
| Undirected | $(1+\varepsilon)$ | 1 | Randomized | [EHW15] |
| | $O(\log n)$ | 1 | Deterministic | Our work |
| | $(2+\varepsilon)$ | $O(\log n)$ | Deterministic | [BKV12] |
| Directed | $\tilde{\Omega}(n^{1/6})$ | 1 | Randomized | [EHW15] |
| | $(2+\varepsilon)$ | 1 | Random order stream | [MP24] |
| | $O(\log n)$ | 1 | Deterministic | Our work |
| | $(2+\varepsilon)$ | $O(\log n)$ | Deterministic | [BKV12] |
| Dynamic model | | | | |
| | Approximation | Memory usage | Update time | Reference |
| Undirected | $(4+\varepsilon)$ | $\tilde{O}(n)$ | $\tilde{O}(1)$ amortized | [BHNT15] |
| | $O(\log n)$ | $\tilde{O}(n)$ | $\tilde{O}(1)$ worst-case | Our work (insertion-only) |
| | $(1+\varepsilon)$ | $\tilde{O}(m)$ | $\tilde{O}(1)$ worst-case | [SW20] |
| Directed | $O(\log n)$ | $\tilde{O}(n)$ | $\tilde{O}(1)$ worst-case | Our work (insertion-only) |
| | $(1+\varepsilon)$ | $\tilde{O}(m)$ | $\tilde{O}(1)$ worst-case | [SW20] |

Table 1: A summary of state-of-the-art results on the DS problem in MPC, semi-streaming, and dynamic models for constants $\epsilon > 0$ and $\delta \in (0,1)$, and graphs with $n$ vertices and $m$ edges.

as memory per computing unit or data access. While it is known how to find a DS in polynomial time [Cha00], it is unclear how to efficiently implement this algorithm in the context of large-scale modern computation. It inspired several research groups to study DS computation in distributed, parallel, and streaming settings under various memory constraints and approximation guarantees.

In Massively Parallel Computation (MPC), which is a theoretical abstraction of popular large-scale frameworks such as MapReduce and Hadoop, [BGM14] proposed an $O(\log n/\varepsilon^2)$ MPC round algorithm for constructing $(1+\varepsilon)$-approximate DS for directed and undirected graphs, where $\varepsilon > 0$ is a precision parameter and $n = |V|$. For undirected DS, this complexity was improved by [GLM19] to $O(\sqrt{\log n} \cdot \log\log n)$ rounds. Recently, in the setting where each machine in MPC has $\tilde{\Theta}(n)$ memory, [MP24] designed an algorithm for $(2+\varepsilon)$-approximation of directed DS that takes $O(\sqrt{\log n})$ rounds.

In the context of semi-streaming, [BKV12] proposed an elegant peeling-based algorithm that computes a $(2+\varepsilon)$-approximation of directed DS in $O(\log n/\varepsilon)$ passes. If the goal is to compute undirected DS, [EHW15] provide a single-pass algorithm that guarantees a $(1+\varepsilon)$-approximation. Interestingly, the same technique for directed DS attains a $\tilde{\Omega}(n^{1/6})$-approximation. Better approximations are only known if the underlying stream is randomized [MP24] or if $\Omega(n^{1.5})$ memory can be used in the semi-streaming setting [EHW15].

Motivated by this disparity between the state-of-the-art results for directed and undirected DS, we ask: *What algorithmic techniques help narrow the gap in computing directed versus undirected DS?*

## 1.1 Our contributions

Table 1 provides a summary of previous state-of-the-art results in comparison to ours.

**Result 1** (Theorem 4.2 rephrased)**.** *Given an $n$-vertex graph and $\varepsilon > 0$, there exists a sublinear MPC algorithm that outputs a $(2+\varepsilon)$-approximate directed DS in $\tilde{O}(\sqrt{\log n})$ rounds. The algorithm uses $O(n^\delta)$ memory per machine and $O(n^{1+\delta} + m)$ total memory for $\delta \in (0,1)$.*

This improves on [BGM14], which uses $O(\log n)$ rounds, and on [MP24], which uses $O(\sqrt{\log n})$ MPC rounds but requires near-linear memory per machine. Additionally, Result 1 matches the state-of-the-art round complexity of $\tilde{O}(\sqrt{\log n})$ for undirected graphs from [GLM19], bridging the gap between the directed and undirected DS problems in MPC.

**Result 2** (Theorem 5.1 rephrased). *Given an $n$-vertex graph and $\varepsilon > 0$, there exists a single-pass deterministic semi-streaming algorithm that outputs an $O(\log n)$-approximate directed DS.*

This is the first single-pass semi-streaming algorithm for the directed DS problem on arbitrary streams. [MP24] attains a $(2 + \varepsilon)$-approximation but only for randomized streams. [EHW15] attains a $(1 + \varepsilon)$-approximation when additional memory, i.e., $O(n^{1.5} \operatorname{poly} \log n)$, is allowed. If we were to extend the ideas of using uniform sampling from prior works, a generalization of the construction in [MP24] shows that it will result in at least a $\tilde{\Omega}(n^{1/6})$-approximation. We also note that this is a deterministic algorithm and can be easily adapted to undirected graphs, leading to the first deterministic single-pass semi-streaming algorithm for the DS problem in both undirected and directed cases.

Result 2 is also an insertion-only dynamic algorithm. Its worst-case update time is $O(\log n)$ for undirected and $O(\log^2 n)$ for directed graphs. To our knowledge, no previous algorithm maintains an approximate *directed* DS in semi-streaming. [BHNT15] shows how to maintain a $(4+\varepsilon)$-approximate undirected DS with $\tilde{O}(1)$ amortized update time and $\tilde{O}(n)$ memory but may have $\Omega(n)$ worst-case update time. [SW20] shows how to maintain a $(1 + \varepsilon)$-approximate directed DS with $O(\log^5 n)$ worst-case update time but requires linear memory.

Empirical evaluation suggests that, in practice, our semi-streaming algorithm yields an approximation much better than $\log n$. On temporal datasets specifically, it matches the approximation of [BKV12].

## 2 Preliminaries

**Transformation: General directed to bipartite undirected graph.** Given a directed graph $G_{\text{dir}} = (V_{\text{dir}}, E_{\text{dir}})$, we represent it as a bipartite graph $G = (V_1, V_2, E)$ where: (i) $V_1$ and $V_2$ are two copies of $V_{\text{dir}}$, and (ii) there is an edge in $E$ between $u \in V_1$ and $v \in V_2$ iff there is a directed edge from $u$ to $v$ in $E_{\text{dir}}$. All directed graphs will be treated as bipartite with this representation.

**Notation.** For a bipartite graph $G = (S, T, E)$, we use $n$ to refer to $|S| + |T|$, the total number of vertices. Given two vertex sets $A \subseteq S$ and $B \subseteq T$, we refer to the edges between them by $E_G(A, B) \overset{\text{def}}{=} \{e = (i, j) \in E : i \in A, j \in B\}$. We use $d_G(v)$ to denote the degree of vertex $v$ in $G$.

**Directed densest subgraph.** Given bipartite graph $G = (V, V, E)$ and vertex sets $S, T \subseteq V$, the *density* $\rho(S, T)$ is defined as $\rho(S, T) \overset{\text{def}}{=} |E_G(S, T)| / \sqrt{|S| \cdot |T|}$. A *directed densest subgraph* is sets $S^*, T^*$ such that $(S^*, T^*) \in \arg\max_{S, T \subseteq V} \rho(S, T)$.

**Massively Parallel Computation (MPC).** In MPC, synchronous rounds of computation are performed across $N$ machines. Each machine has $S$ words of memory and, initially, the input data is arbitrarily distributed across the machines. During a round, each machine computes its local data. Then, after the round, machines exchange messages synchronously. Each machine can send messages to any other machine, but each machine can send and receive at most $S$ words of data. The primary objective is to perform computation in as few rounds as possible. With respect to $S$, three regimes are primarily studied: given $\delta \in (0, 1)$, *sub-linear* ($S = n^\delta$), *near-linear* ($S = n \operatorname{poly} \log n$), and *super-linear* ($S = n^{1+\delta}$). In this work, we focus on the most restrictive sub-linear memory regime. Even though the running time in definition of the MPC model is allowed to be arbitrarily large, the running time of our MPC algorithm is near-linear per round.

**Semi-streaming.** In the semi-streaming setting, an input graph $G = (V, E)$ is given as a stream of edges. That is, an algorithm receives one edge $e \in E$ at a time, and updates its internal memory based on $e$. This internal memory is constrained to be $O(n \cdot \operatorname{poly} \log n)$. After all edges are presented as a stream, we say the algorithm made a *pass* over the graph. An algorithm can make multiple passes over data. During a pass, the algorithm can perform arbitrarily large polynomial-time computations. We remark that our algorithm spends $O(\operatorname{poly} \log n)$ time to update its memory per edge.

# 3 The base algorithm

In this section, we describe the base of our approach. Then, in Sections 4 and 5, we develop new insights enabling us to extend this approach to the MPC and semi-streaming model.

As a reminder, we use $S^*$ and $T^*$ to denote the vertex subsets corresponding to the densest subgraph. Additionally, we view the input graph $G$ as a bipartite undirected rather than a directed graph; details of this transformation are described in Section 2. We assume that our algorithm is given the density $D = \rho(S^*, T^*)$ and the ratio $z = \sqrt{|S^*|/|T^*|}$.[2] Then, leveraging the knowledge of $D$ and $z$, the main idea of this base algorithm is to construct a pair of vertex subsets $(S, T) \subseteq V \times V$ such that it approximates $(S^*, T^*)$ in the following sense:

- Each vertex in $S$ has at least $D/(2z)$ neighbors in $T$.
- Each vertex in $T$ has at least $Dz/2$ neighbors in $S$.

We show that such a pair $(S, T)$ must exist and that it is a 2-approximation of the directed densest subgraph. The proofs of the following lemmas can be found in Appendix A and Appendix B.

**Lemma 3.1** (Subgraph existence). *Let $(S^*, T^*)$ be a directed densest subgraph. Let $D = \rho(S^*, T^*)$ be its density and $z = \sqrt{|S^*|/|T^*|}$. There exists an induced subgraph $H$ on vertex sets $(S, T)$ for which it holds that $d_H(v) \geq D/(2z)$ for all $v \in S$ and $d_H(v) \geq Dz/2$ for all $v \in T$.*

**Lemma 3.2** (Sufficient condition for 2-approximation). *Let $(S^*, T^*)$ be a directed densest subgraph. Let $D = \rho(S^*, T^*)$ be its density and $z = \sqrt{|S^*|/|T^*|}$. Then, any induced subgraph $H$ on vertex sets $(S, T)$ which satisfies $d_H(v) \geq D/(2z)$ for all $v \in S$ and $d_H(v) \geq Dz/2$ for all $v \in T$ has density at least $D/2$. In other words, it is a 2-approximation of the directed densest subgraph.*

Lemma 3.2 provides sufficient conditions under which a given subgraph is a 2-approximate densest one. These conditions inspire a simple peeling procedure for constructing such a subgraph, e.g., Algorithm 1. Let $S$ and $T$ denote the two bipartite sides of that maintained subgraph. Each $S$ and $T$ is initialized to $V$. Then, it iteratively removes vertices that do **not** satisfy the degree conditions stated by Lemma 3.2. More precisely, the algorithm iteratively removes vertices from $S$ with degrees less than $D/(2z)$ and vertices from $T$ with degrees less than $Dz/2$. If **only** these two steps were performed without any extra stopping rule, this algorithm *could execute too many peeling iterations before terminating*. To see that, consider the construction in Figure 1.

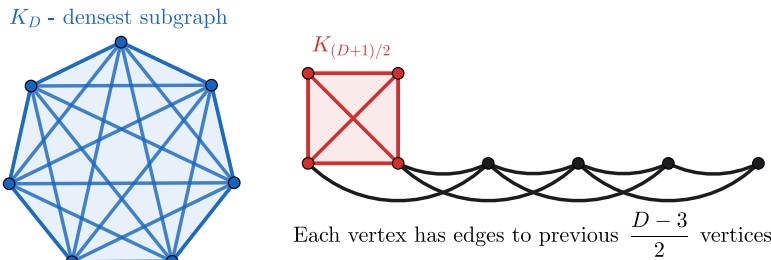

Figure 1: An illustration of the construction that requires $\Omega(n)$ iterations of peeling. We observe that the added vertices (black vertices) can extend as far as desired and will be removed one by one from right to left.

We construct an undirected graph; we obtain its directed version by considering each undirected edge as two directed ones. Let the densest subgraph be the complete graph $K_D$, which has density $D - 1$ and $z = 1$, so we will peel vertices with degree less than $(D - 1)/2$. Then, disjoint to this subgraph, we start with a $\frac{D+1}{2}$-complete subgraph with vertices labeled $1, \ldots, \frac{D+1}{2}$. We continually add vertex $i$ with edges to vertices $i - \frac{D-3}{2} + 1, i - \frac{D-1}{2} + 2, \ldots, i - 1$ for $i > \frac{D+1}{2}$. This disjoint subgraph and all of its subgraphs have a smaller density than $K_D$, and the number of iterations of peeling is linear with the number of vertices we add, resulting in $\Omega(n)$ iterations of peeling.

We make a crucial observation that enables us to avoid this behavior – if a set $S$ or $T$ does not decrease in size by a factor of $(1 + \varepsilon)$ at least, the current $(S, T)$ subgraph already achieves the desired

---

[2]Our final algorithms do not require knowledge of $D$ and $z$. Those details are discussed in Section 4.

approximation (see the proof of Theorem 3.4 for details). This stopping criterion – implemented by Lines 5 and 6 – enables us to ensure sufficient progress in each peeling iteration.

---

**Algorithm 1** Computes a $2(1 + \varepsilon)$-approximate directed densest subgraph assuming that $D = \rho(S^*, T^*)$ and $z = \sqrt{|S^*|/|T^*|}$

---

**Input:** bipartite graph $G = (S, T, E)$, parameters $\varepsilon > 0$, $D$ and $z$
 1: $k_S = D/(2z)$, $k_T = Dz/2$
 2: **while** true **do**
 3:     $A \leftarrow$ all vertices $v \in S$ with $d_G(v) < k_S$
 4:     $B \leftarrow$ all vertices $v \in T$ with $d_G(v) < k_T$
 5:     **if** $|S| \geq z^2|T|$ and $|A| \leq \frac{\varepsilon}{1+\varepsilon}|S|$ **then return** $(S, T)$
 6:     **if** $|S| \leq z^2|T|$ and $|B| \leq \frac{\varepsilon}{1+\varepsilon}|T|$ **then return** $(S, T)$
 7:     Remove $A$ from $S$ and $B$ from $T$

---

### 3.1  Comparison to prior work

The idea of gradually peeling vertices based on their degrees has appeared in prior works on densest subgraphs, e.g., see [BKV12, GLM19, MP24]. Nevertheless, we are unaware of a peeling-based algorithm that uses stopping rules akin to those on Lines 5 and 6 of Algorithm 1. On the one hand, a typical peeling iteration for computing densest subgraphs removes vertices whose degree is less than $1 + \varepsilon$ times the average degree. This immediately implies $O(\log_{1+\varepsilon} n)$ peeling steps and has no need for stopping rules. However, the average degree changes as the graph changes; so, the degree threshold above which to peel vertices evolves. On the other hand, Algorithm 1 uses *the same* degree thresholds, i.e., $k_S$ and $k_T$, throughout the entire execution. Then, the advantage of fixed degree thresholds is that they *do not need to be recomputed* in distributed and streaming settings. This is one of the key properties enabling us to design our algorithms in the coming sections.

### 3.2  Analysis of Algorithm 1

We use *peeling iteration* to refer to a single while-loop iteration of Algorithm 1. During a peeling iteration, all vertices in $S$ or $T$ with degree below the respective threshold are removed, implying:

**Lemma 3.3.** *Algorithm 1 executes $O(\log_{1+\varepsilon} n)$ peeling iterations.*

*Proof.* After a peeling iteration, either $S$ or $T$ decreases in size by a factor of at least $(1 + \varepsilon)$, or the algorithm outputs the current vertex sets and terminates (Lines 5 and 6). Once $S$ and $T$ are empty, Algorithm 1 finishes. Hence, there are $O(\log_{1+\varepsilon} n)$ iterations of peeling. $\square$

**Theorem 3.4.** *Let $(S^*, T^*)$ be a directed densest subgraph of $G$. For $D = \rho(S^*, T^*)$ and $z = \sqrt{|S^*|/|T^*|}$, Algorithm 1 outputs a $2(1 + \varepsilon)$-approximate directed densest subgraph.*

*Proof.* Let $H$ be the subgraph from Lemma 3.1. Since, $D = \rho(S^*, T^*)$ and $z = \sqrt{|S^*|/|T^*|}$, Algorithm 1 will not remove any vertex from $H$. Hence, $S$ and $T$ produced by Algorithm 1 are not empty sets. Therefore, there must exist an iteration of Algorithm 1 in which $A$ (resp. $B$) is at most $\varepsilon/(1+\varepsilon)|S|$ (resp. $\varepsilon/(1+\varepsilon)|T|$). That is, in this iteration, $S$ and $T$ would decrease in size by a factor *less* than $1 + \varepsilon$. Observe that when such an iteration occurs, the algorithm terminates. Therefore, we have the following two cases.

**Case** Algorithm 1 terminates at Line 5. We have that $|A| \leq \frac{\varepsilon}{1+\varepsilon}|S|$ meaning that at least $\frac{1}{1+\varepsilon}|S|$ vertices in $S$ have degree at least $k_S$. This allows us to lower bound the density, giving us

$$\rho(S, T) \geq \frac{k_S \cdot \frac{|S|}{1+\varepsilon}}{\sqrt{|S||T|}} = \frac{D}{2(1+\varepsilon)} \cdot \frac{1}{z}\sqrt{\frac{|S|}{|T|}} \geq \frac{D}{2(1+\varepsilon)}$$

where we used $|S| \geq z^2|T|$. Therefore, the produced subgraph is a $2(1 + \varepsilon)$-approximation.

**Case** Algorithm 1 terminates at Line 6. Similarly, we have that $|B| \leq \frac{\varepsilon}{1+\varepsilon}|T|$ meaning that at least $\frac{1}{1+\varepsilon}|T|$ vertices in $T$ have degree at least $k_T$. This gives us

$$\rho(S, T) \geq \frac{k_T \cdot \frac{|T|}{1+\varepsilon}}{\sqrt{|S||T|}} = \frac{D}{2(1+\varepsilon)} \cdot z\sqrt{\frac{|T|}{|S|}} > \frac{D}{2(1+\varepsilon)}$$

using the $|S| \leq z^2|T|$ constraint. Therefore, this also produces a $2(1 + \varepsilon)$-approximation. $\qquad\square$

# 4 $\tilde{O}(\sqrt{\log n})$ MPC rounds in the sublinear memory regime

In this section, we extend Algorithm 1 to the MPC sublinear memory regime. Directly translating Algorithm 1 into an MPC is straightforward – standard MPC algorithmic primitives, e.g., see [GSZ11, ASS⁺18], enable us to perform one iteration of peeling in $O(1)$ MPC rounds. Since Algorithm 1 takes $O(\log n)$ iterations, this direct MPC implementation results in an $O(\log n)$ MPC round algorithm in the sublinear memory regime. However, we aim to obtain a quadratically faster algorithm.

## 4.1 Our improved approach

On a high level, we execute the $O(\log n)$ iterations of (a variant of) Algorithm 1 in $\tilde{O}(\sqrt{\log n})$ MPC rounds. We present the intuition behind this approach in two steps: (i) Recall a known framework for simulating $O(\log n)$ iterations in $\tilde{O}(\sqrt{\log n})$ MPC rounds; and (ii) Describe a modified variant of Algorithm 1 that fits into that simulation framework.

**Known simulation techniques.** Consider a $T$-iteration algorithm $\mathcal{A}_{\text{LOCAL}}$ such that: (1) $\mathcal{A}_{\text{LOCAL}}$ maintains a state of each vertex, e.g., a vertex $v$ is removed or not, and (2) the state of $v$ in iteration $i$ depends only on the states of $v$ and its neighbors in iteration $i-1$, e.g., $\mathcal{A}_{\text{LOCAL}}$ decides whether $v$ should be removed in iteration $i$ based on the number of $v$'s non-removed neighbors in iteration $i-1$. The **output** of $\mathcal{A}_{\text{LOCAL}}$ is the state of each vertex in each of the $T$ iterations.

*Remark*: Algorithm 1 does not have the properties of $\mathcal{A}_{\text{LOCAL}}$, e.g., evaluating the conditions on Lines 5 and 6 requires "global" computation. However, we note that these lines can be simulated through post-processing, constructing sets $S$ and $T$ for each iteration and comparing their sizes using the states of vertices. The remaining steps of Algorithm 1 can be phrased in the language of $\mathcal{A}_{\text{LOCAL}}$.

*In how many MPC rounds can we execute $\mathcal{A}_{\text{LOCAL}}$?* Based on our description, observe that the state of a vertex $v$ at any moment during the execution of $\mathcal{A}_{\text{LOCAL}}$ is a function of the $v$'s $T$-hop neighborhood. Let $N_j(v)$ denote the $j$-hop neighborhood of $v$. This observation gives rise to the following idea:

(1) Gather $N_T(v)$ of each vertex $v \in V$.

(2) Place $N_T(v)$ on a single machine in MPC; $N_T(v)$ for different $v$ are placed on different machines.

(3) In a single MPC round, execute $\mathcal{A}_{\text{LOCAL}}$ within $N_T(v)$ to learn the state of $v$.

Assuming that $N_T(v)$ fits in the memory of a single machine, a significant advantage of this approach is that it takes only $O(\log T)$ MPC rounds. Namely, Item 1 can be implemented using a well-known technique *graph exponentiation* [LW10, Gha17]. In this technique, $N_{2^i}(v)$ is gathered for each $v \in V$ and for each $i = 0 \cdots \log T$, using the following relation:

$$N_{2^{i+1}}(v) = \bigcup_{w \in N_{2^i}(v)} N_{2^i}(w).$$

Namely, each iteration of the graph exponentiation technique doubles the radius of the neighborhood collected around a vertex. To implement this technique in MPC, it is necessary to address: *Can $N_T(v)$ fit into the memory of a single machine in MPC?* This takes us to the second part of our approach.

**Graph sparsification (Algorithm 2) – A modified variant of Algorithm 1.** We can alter the recipe above to simulate $\mathcal{A}_{\text{LOCAL}}$ in MPC while still aiming for $o(T)$ rounds. Namely, (i) we can split those $T$ iterations into $T/k$ groups, each consisting of $k$ consecutive iterations, (ii) execute group after group sequentially, such that (iii) each group is executed as described above by using graph exponentiation. Setting aside memory constraints, this method uses $O\left(\frac{T}{k} \cdot \log k\right)$ MPC rounds. The larger $k$ is, the fewer MPC rounds are needed. However, a larger $k$ implies a bigger $N_k(v)$ which might not fit in a

---

**Algorithm 2** Finds partial $(2 + \varepsilon)$-approximation of directed densest subgraph

---

**Input:** bipartite graph $G = (S, T, E)$, $\varepsilon \in (0, 1)$, $\delta \in (0, 1)$, $D$ and $z$

1: $k_S = D/(2z)$, $k_T = Dz/2$, $\alpha = (1 + \varepsilon)^{\sqrt{\log_{1+\varepsilon} n}}$
2: Freeze all vertices in $S$ of degree greater than $k_S \alpha$
3: Freeze all vertices in $T$ of degree greater than $k_T \alpha$
4: Mark as frozen each edge with both endpoints frozen
5: $f_1 \leftarrow$ number of frozen vertices in $S$
6: $f_2 \leftarrow$ number of frozen vertices in $T$
7: **if** $f_1 \geq \frac{z\sqrt{|S||T|}}{\alpha}$ or $f_2 \geq \frac{\sqrt{|S||T|}}{z\alpha}$ **then return** $(S, T)$
8: $p_1 \leftarrow \min\left(1, \frac{18 \log n}{\varepsilon^2 k_S}\right)$
9: $p_2 \leftarrow \min\left(1, \frac{18 \log n}{\varepsilon^2 k_T}\right)$
10: $t \leftarrow \frac{\sqrt{\delta \log_{1+\varepsilon} n}}{2}$
11: **for** $t$ steps **do**
12:     $G_1 \leftarrow$ sample of each non-frozen edge of $G$ with probability $p_1$
13:     $G_2 \leftarrow$ sample of each non-frozen edge of $G$ with probability $p_2$
14:     $A \leftarrow$ all non-frozen vertices $v \in S$ with $d_{G_1}(v) < pk_S$
15:     $B \leftarrow$ all non-frozen vertices $v \in T$ with $d_{G_2}(v) < pk_T$
16:     **if** $|S| \geq z^2 |T|$ and $|A| \leq \frac{\varepsilon}{1+\varepsilon}|S| - f_1$ **then return** $(S, T)$
17:     **if** $|S| \leq z^2 |T|$ and $|B| \leq \frac{\varepsilon}{1+\varepsilon}|T| - f_2$ **then return** $(S, T)$
18:     Remove $A$ from $S$ and $B$ from $T$
19: **return** $(S, T)$

---

machine's memory. It turns out that $k = \Theta\left(\sqrt{\log n}\right)$ is the largest $k$ our approach can tolerate after performing certain graph sparsification from [GU19] described next.

Fix $t = \Theta(\sqrt{\log n})$. Ignoring the memory-per-machine constraint, $N_t(v)$ can be collected in $O(\log \log n)$ MPC rounds. However, we cannot guarantee that these neighborhoods fit on a single machine with $O(n^\delta)$ memory, as some vertices can have large degrees. For example, vertices with a degree of $\omega(n^\delta)$ cannot even store their entire neighborhood on a single machine. Inspired by this, Algorithm 2 temporarily "ignores" these high-degree vertices when performing graph exponentiation. We call this process *freezing* high-degree vertices, seen on Line 2 and Line 3. Intuitively, freezing enables us to transform the current graph into one having sufficiently small degrees, and hence the graph exponentiation can be executed with $O\left(n^\delta\right)$ memory per machine. On Line 4, we also ignore the edges between frozen vertices since they do not affect the peeling of non-frozen vertices.

Algorithm 2 samples the graph on top of freezing high-degree vertices on Line 12 and Line 13. Combining both freezing and sampling, the graph becomes sparse enough so that $\Theta(t)$-hop neighborhoods fit within sublinear memory and peeling is simulated with high probability. Therefore, if $\Theta(t)$ iterations of peeling can be simulated in $O(\log \log n)$ rounds, then we can simulate the entirety of Algorithm 1 in $O(\sqrt{\log n} \cdot \log \log n)$ rounds. Between these phases of simulating $\Theta(\sqrt{\log n})$ iterations of peeling, frozen vertices are updated based on their new degrees. However, these frozen vertices are not peeled during these phases, and therefore our peeling does not quite match $\Theta(\sqrt{\log n})$ iterations of peeling in Algorithm 1. Nevertheless, our final algorithm uses $\tilde{O}(\sqrt{\log n})$ rounds. We show that the fraction of frozen vertices is too small to affect the entire peeling process.

### 4.2 Analysis of Algorithm 2

When trying to simulate Algorithm 1, it is important to note that our stopping rules are affected since we have limited information on the degrees of frozen vertices. So, we weaken our rules and this is reflected in the differences between Line 5 and Line 6 of Algorithm 1 and Line 16 and Line 17 of Algorithm 2. Algorithm 2 simulates $t = \Theta(\sqrt{\log n})$ iterations of peeling as described above, but due to frozen vertices, it is not obvious how much the sizes of vertex sets decrease by. Nevertheless, we establish the following claim about the number of frozen vertices. The proof of Lemma 4.1 is deferred to Appendix C.

**Lemma 4.1.** *Let $G = (S, T, E)$ be a bipartite graph and $(S^*, T^*)$ be its directed densest subgraph. Let $(S', T')$ be the output of Algorithm 2 ran on $G$ given $\frac{\rho(S^*, T^*)}{(1+\varepsilon)^3} \leq D \leq \frac{\rho(S^*, T^*)}{(1+\varepsilon)^2}$ and $\frac{\sqrt{|S^*|/|T^*|}}{(1+\varepsilon)} \leq z \leq \sqrt{|S^*|/|T^*|}$. Then, with probability at least $1 - \frac{t}{n^2}$, it holds that:*

- ***Good approximation.** $(S', T')$ is a $2(1+\varepsilon)^6$-approximate densest subgraph of $G$, or*
- ***Size reduction.** $(S', T')$ contains a $2(1+\varepsilon)^6$-approximate densest subgraph of $G$ and*

$$|S'| \leq \frac{|S|}{\gamma} \text{ or } |T'| \leq \frac{|T|}{\gamma}$$

*where $\gamma = (1+\varepsilon)^{\frac{\sqrt{\delta \log_{1+\varepsilon} n}}{8}}$.*

Following the ideas described before, we invoke Algorithm 2 $\Theta(\sqrt{\log n})$ times to simulate all iterations of peeling, resulting in the following theorem, whose proof is deferred to Appendix D.

**Theorem 4.2.** *There exists a sublinear MPC algorithm that runs in $\tilde{O}(\sqrt{\log n})$ rounds and attains a $2(1+\varepsilon)^6$-approximation of the directed densest subgraph with probability at least $1 - \frac{1}{n}$. The algorithm uses $O(n^\delta)$ memory per machine and $O(n^{1+\delta} + m)$ total memory for $\delta \in (0, 1)$.*

## 5 Single-pass semi-streaming algorithm

In this section, we extend Algorithm 1 to the single-pass semi-streaming setting. It is unclear how to adapt the sampling used for the MPC algorithm since the number of edges sampled could be $\omega(n \operatorname{poly} \log n)$, surpassing the memory limit of the semi-streaming model. Nonetheless, we show that an $O(\log n)$-approximate directed densest subgraph can be obtained by maintaining **only vertex degrees** throughout the stream. This directly results in a $\tilde{O}(n)$ memory requirement, as a vertex's degree can be maintained using a simple integer counter. To the best of our knowledge, this is the first semi-streaming single-pass algorithm for the directed densest subgraph problem that achieves better than $\operatorname{poly} n$ approximation.

### 5.1 Our approach

---

**Algorithm 3** Finds $O(\log n)$-approximation of directed densest subgraph

---

**Input:** bipartite graph $G = (S, T, E), \varepsilon > 0, D$ and $z$
1: $k_S = D/(2z), k_T = Dz/2$
2: $l_S(v), l_T(v) \leftarrow 0$ for all $v \in V$          ▷ $l$ stands for level
3: $d_S(v), d_T(v) \leftarrow 0$ for all $v \in V$      ▷ $d$ stands for vertex degree in **its current** level
4: **while** stream not empty **do**
5:      $(u, v) \leftarrow$ next edge from stream
6:      **if** $l_S(u) \leq l_T(v)$ **then** $d_S(u) \leftarrow d_S(u) + 1$
7:      **if** $l_S(u) \geq l_T(v)$ **then** $d_T(v) \leftarrow d_T(v) + 1$
8:      **if** $d_S(u) \geq k_S$ **then**
9:          $l_S(u) \leftarrow l_S(u) + 1$
10:        $d_S(u) \leftarrow 0$
11:      **if** $d_T(v) \geq k_T$ **then**
12:        $l_T(v) \leftarrow l_T(v) + 1$
13:        $d_T(v) \leftarrow 0$
14: $S_i \leftarrow \{v : l_S(v) \geq i\}$ for all $0 \leq i \leq 2 \log_{1+\varepsilon} n$
15: $T_i \leftarrow \{v : l_T(v) \geq i\}$ for all $0 \leq i \leq 2 \log_{1+\varepsilon} n$
16: **for** $i = 1 \ldots 2 \log_{1+\varepsilon} n$ **do**
17:      **if** $|S_i| \geq z^2 |T_i|$ and $|S_i| \geq \frac{|S_{i-1}|}{1+\varepsilon}$ **then return** $(S_i, T_i)$
18:      **if** $|S_i| \leq z^2 |T_i|$ and $|T_i| \geq \frac{|T_{i-1}|}{1+\varepsilon}$ **then return** $(S_i, T_i)$

---

*How can vertex degrees be leveraged to (approximately) simulate Algorithm 1?* We observe that Algorithm 1 implicitly computes a vertex-vector $l$, where $l(v)$ is the peeling iteration after which

vertex $v$ was removed from the graph. We refer to $l(v)$ as the *level* of $v$. Having $l$ suffices to recover $(S, T)$ in each iteration of the while-loop. Our approach approximates $l(v)$ for each vertex by using the evolution of vertex degrees as the stream is read. We now discuss details.

As a reminder, we think of an input graph as bipartite with the bipartitions $S$ and $T$; details are described in Section 2. As our algorithm scans the edges in a stream, the maintained vertex degrees increase. Initially, each vertex level and vertex degree is $0$. A vertex's level $l(v)$ increases when the algorithm is certain that $v$ will not be deleted from the graph within the first $l(v)$ peeling iterations.

On a high level, once a vertex degree reaches $D/(2z)$ if it is in $S$ or $Dz/2$ if it is in $T$, it will not be removed during the first peeling iteration. Therefore, we can confidently increase its level to 1. For a vertex in $S$ to move to a level of at least 2, its degree must be at least $D/(2z)$ after the 1st peeling iteration. Unfortunately, before reading the entire stream, our single-pass algorithm cannot say with certainty that a vertex will be peeled in the 1st peeling iteration. However, if a vertex $v$ is **not** peeled in the 1st iteration, an algorithm learns that information about $v$ potentially before reading the entire stream. Inspired by this, we estimate the levels of vertices as follows.

For an edge $(u, v)$ on the stream, we increase $d(u)$ only if $l(v) \geq l(u)$. This guarantees we only count edges where the other vertex has not been removed yet. Once a vertex moves to a higher level, we need to determine its degree at that level. However, it is not obvious how to calculate this degree without maintaining the edges in the subgraphs of each level. Therefore, in Algorithm 3, we assume the worst case and reset the degree of the vertex to $0$ when it moves to a higher level. This poses challenges in analyzing whether these degree estimates are accurate enough to output an approximation of the directed densest subgraph, especially when the stream of edges is adversarial. Fortunately, we show that Algorithm 3 ran for specific input of $D$ and $z$ produces an $O(\log n)$-approximation. Precisely, we show the following result, whose proof is deferred to Appendix E.

**Theorem 5.1.** *There exists a single-pass semi-streaming algorithm that attains an $O(\log n)$-approximation of the directed densest subgraph.*

*Remark.* To adapt the algorithm to finding the *undirected* densest subgraph, we no longer need $z$. All other ideas remain the same, resulting in a single-pass semi-streaming algorithm for approximating the undirected densest subgraph as well. We also have a dynamic algorithm with $O(1)$ update time for each guess on $D$ and $z$. We can maintain sets $(S_i, T_i)$ throughout the algorithm without increasing the update time, resulting in $O(\log n)$ update time and $O(\log^2 n)$ update time for directed graphs. We can output our approximation of the densest subgraph anytime in $O(\text{poly} \log n)$ time.

## 6   Experiments

**Baselines.** We empirically evaluate the performance of our semi-streaming algorithm, comparing it to the $(2 + \varepsilon)$-approximation $O(\log n)$ pass semi-streaming algorithm from [BKV12]. We also compare it to the single-pass semi-streaming algorithm from [MP24], but the density plots of [MP24] follow very similarly to [BKV12]. So, we only include [BKV12] in our approximation plots in Figure 2. As discussed in the introduction, [MP24] can have a quite high worst-case update time per edge on a stream, while the update time of our work is $O(\log^2 n)$. We illustrate this empirically in Figure 3.

To run these experiments, we use an M1 machine running macOS Sequoia 15.3.2 with 4 cores, 8 GB of RAM, 256 KB of L2 Cache, and 2.5 MB of L3 Cache (per core).

**Data.** We use 10 datasets from the Stanford Large Network Dataset Collection [LK14]. 5 of these datasets (Slashdot, Berkeley-Stanford Web, Google Web, Pokec, LiveJournal) are general directed graphs; most consist of edges sorted by endpoints. The other 5 datasets (Ask Ubuntu, Super User, Wikipedia, Twitter, Stack Overflow) are temporal directed graphs where edges are in sorted time order of when interactions happened. These temporal graphs reflect a stream of edge updates.

**Results.** We run the algorithms on temporal directed graphs with $\varepsilon = 0.2$. These datasets present events/edges in the order they occurred, making them an excellent benchmark for practical applications of dynamic algorithms. In Figure 2, we see that our algorithm matches [BKV12] for the largest density. We also observe that it is significantly less sensitive to error in $z^2$, making it better in practice than [BKV12] when our approximation of $z^2$ may not be as precise. Overall, we demonstrate that in practice our algorithm performs much better than the $\log n$ theoretical guarantee would imply.

| Graph | Nodes | Edges |
|---|---|---|
| Slashdot | 82,168 | 948,464 |
| Google Web | 875,713 | 5,105,039 |
| Berk-Stan Web | 685,230 | 7,600,595 |
| Pokec | 1,632,803 | 30,622,564 |
| LiveJournal | 4,847,571 | 68,993,773 |

| Graph | Nodes | Edges |
|---|---|---|
| Ask Ubuntu | 159,316 | 964,437 |
| Super User | 194,085 | 1,443,339 |
| Wikipedia | 1,140,149 | 7,833,140 |
| Twitter | 456,631 | 14,855,875 |
| Stack Overflow | 2,601,977 | 63,497,050 |

(a) General directed graphs    (b) Temporal directed graphs

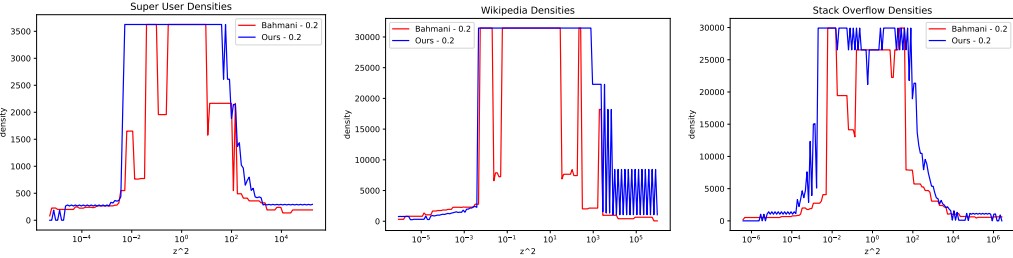

Figure 2: Density as a function of $z^2$ for various temporal datasets.

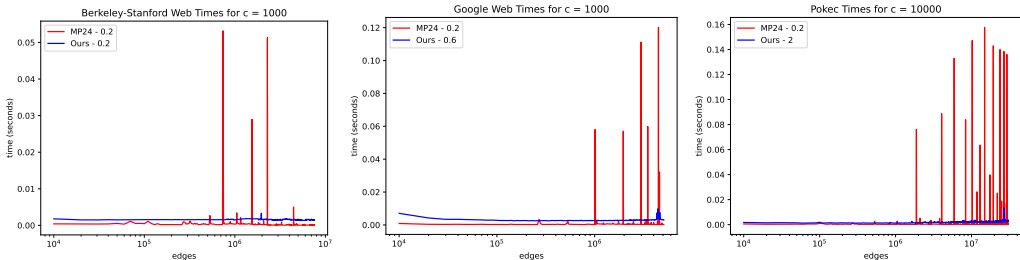

Figure 3: Update time between processing edges of the stream for various general datasets.

We also compare the update time of our algorithm to [MP24], using their threshold $f = 1/450$, in Figure 3. The plot contains update times of the algorithms for batches of 10,000 edges. The chosen $c$ are where the largest densities are attained, and the chosen $\varepsilon$ are based on maximizing densities. Note that $\varepsilon$ does not significantly affect the running time of our algorithm. As we can see, the worst-case update time of our algorithm is significantly more stable and lower compared to [MP24].

Experimental results for the remaining datasets can be found in Appendix F. Additionally, we provide experiments for our MPC algorithm in Appendix G which show that our algorithm not only improves the theoretical upper bound on round complexity but also uses fewer rounds in practice.

## 7 Conclusion and future work

We study the directed densest subgraph problem in MPC and semi-streaming models. Our MPC algorithm bridges the gap between known algorithms for computing undirected and directed approximate DS. We also develop a simple deterministic single-pass semi-streaming algorithm. This is the first single-pass algorithm for the directed DS problem to achieve a sub-polynomial approximation.

Our work leaves a few intriguing questions. First, even though our MPC algorithm is able to match the round complexity of the state-of-the-art undirected DS algorithm, there is still a gap between their approximation factors, $1 + \varepsilon$ compared to $2 + \varepsilon$. Is it possible to improve the approximation guarantee for the directed DS problem while not increasing the round complexity? Second, can this upper bound of $\tilde{O}(\sqrt{\log n})$ MPC rounds in the sublinear memory regime be broken for either undirected or directed graphs? Third, our semi-streaming algorithm attains an $O(\log n)$-approximation. Can we develop a single-pass semi-streaming algorithm with an $\Theta(1)$-approximation for directed graphs?

## Acknowledgments and Disclosure of Funding

S. Mitrović and T. Pan were supported by NSF CAREER award, No. 2340048. We are grateful to anonymous reviewers for their valuable feedback.

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

# A Proof of Lemma 3.1

**Lemma 3.1** (Subgraph existence). *Let $(S^*, T^*)$ be a directed densest subgraph. Let $D = \rho(S^*, T^*)$ be its density and $z = \sqrt{|S^*|/|T^*|}$. There exists an induced subgraph $H$ on vertex sets $(S, T)$ for which it holds that $d_H(v) \geq D/(2z)$ for all $v \in S$ and $d_H(v) \geq Dz/2$ for all $v \in T$.*

*Proof.* We claim that the directed densest subgraph satisfies the constraints of such a subgraph $H$. Consider any vertex $v \in S^*$. Since $H$ is the directed densest subgraph, removing $v$ does not increase the density. Therefore, we have

$$
\begin{aligned}
\frac{|E_G(S^*, T^*)|}{\sqrt{|S^*||T^*|}} &\geq \frac{|E_G(S^*, T^*)| - d_H(v)}{\sqrt{(|S^*| - 1)|T^*|}} \\
\implies d_H(v) &\geq \left(1 - \sqrt{\frac{|S^*| - 1}{|S^*|}}\right)|E_G(S^*, T^*)| \\
\implies d_H(v) &\geq \frac{|E_G(S^*, T^*)|}{2|S^*|} = \frac{D}{2z}
\end{aligned}
$$

where for the last inequality, we used $(1 - 1/|S^*|)^{1/2} \leq 1 - 1/(2|S^*|)$. Similarly, we can show that for any vertex $v \in T^*$, it must satisfy $d_H(v) \geq Dz/2$.

Hence, the pair $(S^*, T^*)$ satisfies the constraints of the claim. $\qquad\square$

# B Proof of Lemma 3.2

**Lemma 3.2** (Sufficient condition for 2-approximation). *Let $(S^*, T^*)$ be a directed densest subgraph. Let $D = \rho(S^*, T^*)$ be its density and $z = \sqrt{|S^*|/|T^*|}$. Then, any induced subgraph $H$ on vertex sets $(S, T)$ which satisfies $d_H(v) \geq D/(2z)$ for all $v \in S$ and $d_H(v) \geq Dz/2$ for all $v \in T$ has density at least $D/2$. In other words, it is a 2-approximation of the directed densest subgraph.*

*Proof.* The amount of edges in $H$ is lower bounded by $\max\left(|S| \cdot \frac{D^*}{2z}, |T| \cdot \frac{D^*z}{2}\right)$. Therefore, we have

$$
\begin{aligned}
\rho(H) &\geq \frac{\max\left(|S| \cdot \frac{D^*}{2z}, |T| \cdot \frac{D^*z}{2}\right)}{\sqrt{|S||T|}} \\
&= \frac{D^*}{2} \cdot \max\left(\frac{1}{z}\sqrt{\frac{|S|}{|T|}}, z\sqrt{\frac{|T|}{|S|}}\right) \\
&\geq \frac{D^*}{2}
\end{aligned}
$$

since the two parameters in $\max\left(\frac{1}{z}\sqrt{\frac{|S|}{|T|}}, z\sqrt{\frac{|T|}{|S|}}\right)$ are reciprocals of each other, implying one of them is at least 1. $\qquad\square$

# C Proof of Lemma 4.1

We will use the following lemma to connect degrees between the original graph and the sampled graph.

**Lemma C.1.** *Consider graph $G$, $k > 0$, $\varepsilon \in (0, 1)$. Let $H$ be a subgraph of $G$ obtained by sampling each edge of $G$ independently with probability $p = \min\left(1, \frac{18 \log n}{\varepsilon^2 k}\right)$. Then with probability at least $1 - \frac{1}{n^2}$, for all $v \in G$, it holds:*

- *if $d_G(v) \leq \frac{k}{1+\varepsilon}$, then $d_H(v) < pk$;*

- *if $d_G(v) \geq (1 + \varepsilon)k$, then $d_H(v) > pk$.*

*Proof.* If $p = 1$, then the lemma follows immediately. So, let $p = \frac{18 \log n}{\varepsilon^2 k}$. Consider any vertex $v \in G$. If $d_G(v) \leq k/(1 + \varepsilon)$, then we consider two cases. When $d_G(v) \leq k/2$, we have

$$
\begin{aligned}
\Pr[d_H(v) \geq pk] &\leq \exp\left(-\left(\frac{pk}{\mathbb{E}\left[d_H(v)\right]} - 1\right)\mathbb{E}\left[d_H(v)\right]/3\right) \\
&= \exp\left(-\frac{pk}{3} + \frac{\mathbb{E}\left[d_H(v)\right]}{3}\right) \\
&\leq \exp\left(-pk/6\right) \leq \frac{1}{n^3}
\end{aligned}
$$

and when $k/2 < d_G(v) \leq k/(1 + \varepsilon)$, we have

$$
\begin{aligned}
\Pr[d_H(v) \geq pk] &\leq \exp\left(-\left(\frac{pk}{\mathbb{E}\left[d_H(v)\right]} - 1\right)^2\mathbb{E}\left[d_H(v)\right]/3\right) \\
&\leq \exp\left(-\varepsilon^2 pk/6\right) \leq \frac{1}{n^3}.
\end{aligned}
$$

Therefore, using the union bound over all vertices, we have that $d_H(v) < pk$ for all $v \in G$, satisfying $d_G(v) \leq k/(1 + \varepsilon)$, with probability at least $1 - \frac{1}{n^2}$. Now, if $d_G(v) \geq (1 + \varepsilon)k$, then we have

$$
\begin{aligned}
\Pr[d_H(v) \leq pk] &\leq \exp\left(-\left(1 - \frac{pk}{\mathbb{E}\left[d_H(v)\right]}\right)^2\mathbb{E}\left[d_H(v)\right]/2\right) \\
&\leq \exp\left(-\left(\frac{\varepsilon}{1 + \varepsilon}\right)^2(1 + \varepsilon)pk/2\right) \\
&\leq \exp\left(-\varepsilon^2 pk/6\right) \leq \frac{1}{n^3}.
\end{aligned}
$$

Similarly, using the union bound over all vertices, we have that $d_H(v) > pk$ for all $v \in G$, satisfying $d_G(v) \geq (1 + \varepsilon)k$, with probability at least $1 - \frac{1}{n^2}$. $\qquad\square$

**Lemma 4.1.** *Let $G = (S, T, E)$ be a bipartite graph and $(S^*, T^*)$ be its directed densest subgraph. Let $(S', T')$ be the output of [Algorithm 2](#) ran on $G$ given $\frac{\rho(S^*, T^*)}{(1 + \varepsilon)^3} \leq D \leq \frac{\rho(S^*, T^*)}{(1 + \varepsilon)^2}$ and $\frac{\sqrt{|S^*|/|T^*|}}{(1 + \varepsilon)} \leq z \leq \sqrt{|S^*|/|T^*|}$. Then, with probability at least $1 - \frac{t}{n^2}$, it holds that:*

- ***Good approximation.*** *$(S', T')$ is a $2(1 + \varepsilon)^6$-approximate densest subgraph of $G$, or*

- ***Size reduction.*** *$(S', T')$ contains a $2(1 + \varepsilon)^6$-approximate densest subgraph of $G$ and*

$$
|S'| \leq \frac{|S|}{\gamma} \text{ or } |T'| \leq \frac{|T|}{\gamma}
$$

*where $\gamma = (1 + \varepsilon)^{\frac{\sqrt{\delta \log_{1+\varepsilon} n}}{8}}$.*

*Proof.* Without loss of generality, assume the inputs of [Algorithm 2](#) satisfy $\frac{|S|}{|T|} = cz^2$ for some $c > 1$. Now, if $f_1 \geq \frac{z\sqrt{|S||T|}}{\alpha}$, we use the high degrees of frozen vertices to see that

$$
\rho(G) \geq \frac{f_1 k_S \alpha}{\sqrt{|S||T|}} \geq \frac{D}{2} \geq \frac{\rho(S^*, T^*)}{2(1 + \varepsilon)^3}
$$

meaning that $G$ is a $2(1 + \varepsilon)^3$-approximation of the densest subgraph. A similar argument can be made if $f_2 \geq \frac{\sqrt{|S||T|}}{z\alpha}$, so we assume that both of these conditions are not satisfied.

Now, if we look at the peeling process, it closely simulates Algorithm 1. Using Lemma C.1 and the constraints on $D$ and $z$, we see that we're removing vertices in $S$ with degree below $k_S/(1+\varepsilon)$ and keeping vertices in $S$ with degree at least $(1+\varepsilon)k_S$ where

$$\frac{\rho(S^*,T^*)}{2(1+\varepsilon)^3\sqrt{|S^*||T^*|}} \leq k_S \leq \frac{\rho(S^*,T^*)}{2(1+\varepsilon)\sqrt{|S^*||T^*|}}$$

with probability at least $1 - \frac{t}{n^2}$, using the union bound over all $t$ iterations of peeling. Similarly, we see that we're removing vertices in $T$ with degree below $k_T/(1+\varepsilon)$ and keeping vertices in $T$ with degree at least $(1+\varepsilon)k_T$ where

$$\frac{\rho(S^*,T^*)}{2(1+\varepsilon)^4\sqrt{|S^*||T^*|}} \leq k_T \leq \frac{\rho(S^*,T^*)}{2(1+\varepsilon)^2\sqrt{|S^*||T^*|}}$$

also with probability at least $1 - \frac{t}{n^2}$. Following the same argument behind Theorem 3.4 but using these bounds on $k_S$ and $k_T$, we see that returning early will result in a $2(1+\varepsilon)^6$-approximation in total. Therefore, we will assume Algorithm 2 does not return early. Then, we consider two cases.

If $c \leq (1+\varepsilon)^{\sqrt{\delta \log_{1+\varepsilon} n}}$, then we see that

$$f_1 < \frac{z\sqrt{|S||T|}}{\alpha} \leq \frac{|S|}{(1+\varepsilon)^{\sqrt{\delta \log_{1+\varepsilon} n}}}$$

and

$$f_2 < \frac{\sqrt{|S||T|}}{z\alpha} \leq \frac{|T|}{(1+\varepsilon)^{\sqrt{\delta \log_{1+\varepsilon} n}/2}}.$$

Therefore, after $t$ iterations of peeling, we have that the final returned vertex sets $(S',T')$ satisfy

$$|S'| \leq \frac{|S|-(1+\varepsilon)f_1}{(1+\varepsilon)^{\lfloor t/2 \rfloor}} + (1+\varepsilon)f_1 \leq \frac{|S|}{(1+\varepsilon)^{\sqrt{\delta \log_{1+\varepsilon} n}/8}}$$

or

$$|T'| \leq \frac{|T|-(1+\varepsilon)f_2}{(1+\varepsilon)^{\lfloor t/2 \rfloor}} + (1+\varepsilon)f_2 \leq \frac{|T|}{(1+\varepsilon)^{\sqrt{\delta \log_{1+\varepsilon} n}/8}}$$

because of Line 16 and Line 17, using the upper bounds on $f_1$ and $f_2$, and sufficiently large $n$.

On the other hand, if $c > (1+\varepsilon)^{\sqrt{\delta \log_{1+\varepsilon} n}}$, then we still see that $f_1 \leq \frac{|S|}{(1+\varepsilon)^{\sqrt{\delta \log_{1+\varepsilon} n}}}$. So, if $|S|/|T| \geq z^2$ remains true throughout the whole algorithm, then due to Line 16 we have that the final returned vertex sets $(S',T')$ satisfy

$$|S'| \leq \frac{|S|-(1+\varepsilon)f_1}{(1+\varepsilon)^t} + (1+\varepsilon)f_1 \leq \frac{|S|}{(1+\varepsilon)^{\sqrt{\delta \log_{1+\varepsilon} n}/8}}$$

following a similar argument as above. Otherwise, at some point we have that $|S|/|T| < z^2$, meaning that $|S|$ must have decreased by at least a factor of $c$. As a result, we also have that $|S'| \leq \frac{|S|}{(1+\varepsilon)^{\sqrt{\delta \log_{1+\varepsilon} n}/8}}$ in this case due to the lower bound on $c$. $\qquad\square$

## D  Proof of Theorem 4.2

**Theorem 4.2.** *There exists a sublinear MPC algorithm that runs in $\tilde{O}(\sqrt{\log n})$ rounds and attains a $2(1+\varepsilon)^6$-approximation of the directed densest subgraph with probability at least $1 - \frac{1}{n}$. The algorithm uses $O(n^\delta)$ memory per machine and $O(n^{1+\delta}+m)$ total memory for $\delta \in (0,1)$.*

*Proof.* Let $\mathcal{A}_{\mathrm{MPC}}$ be our MPC algorithm. We first describe $\mathcal{A}_{\mathrm{MPC}}$ and then provide its analysis.

**Algorithm description.** $\mathcal{A}_{\text{MPC}}$ considers all guesses on $1 \leq D \leq n$ and $\frac{1}{\sqrt{n}} \leq z \leq \sqrt{n}$ in parallel using powers of $1 + \varepsilon$. For each of these parallel instances of $D$ and $z$,

- The algorithm invokes Algorithm 2 $16\sqrt{\delta \log_{1+\varepsilon} n}/\delta$ times, constantly providing back the output of the previous invocation, with the edges of the induced subgraph of the new vertex sets, into the new invocation.

- If Algorithm 2 ever returns early in any of these invocations, then we call the pair of vertex sets that is returned early a pair of *potential vertex sets*.

$\mathcal{A}_{\text{MPC}}$ outputs the pair of potential vertex sets with the largest density over all instances of $D$ and $z$.

**Algorithm memory and round complexity.** Using the same application of the graph exponentiation technique shown in [GLM19, Appendix C.2], we see that each machine will only use $O(n^\delta)$ memory since neighborhoods will have size bounded by

$$\left( \frac{36 \log n}{\varepsilon^2} \cdot \alpha \right)^t \in O(n^\delta)$$

with high probability and Algorithm 2 will take $O(\log \log n)$ rounds. Including the number of invocations to Algorithm 2, we have that $\mathcal{A}_{\text{MPC}}$ takes $O(\sqrt{\log n} \cdot \log \log n)$ rounds. Running the algorithm in parallel over all instances of $D$ and $z$ adds an $O(\log^2 n)$ factor to the memory per machine.

**Algorithm approximation.** Note that there will be guesses of $D$ and $z$ that satisfy

$$\frac{\rho(S^*, T^*)}{(1 + \varepsilon)^3} \leq D \leq \frac{\rho(S^*, T^*)}{(1 + \varepsilon)^2} \text{ and } \frac{\sqrt{|S^*|/|T^*|}}{(1 + \varepsilon)} \leq z \leq \sqrt{|S^*|/|T^*|}.$$

Using these guesses and the approximation guarantees from Lemma 4.1, we guarantee these potential vertex sets to be a $2(1 + \varepsilon)^6$-approximation with at least probability $1 - \frac{1}{n}$, taking the union bound over the invocations of Algorithm 2. Since $\mathcal{A}_{\text{MPC}}$ selects the vertex sets with the largest density, it attains at least a $2(1 + \varepsilon)^6$-approximation of the directed densest subgraph. $\qquad\square$

## E  Proof of Theorem 5.1

**Lemma E.1.** *Let the directed densest subgraph have vertex sets $(S^*, T^*)$. Then, $(S_i, T_i)$ in Algorithm 3 will be non-empty vertex sets for all $0 \leq i \leq 2\log_{1+\varepsilon} n$ given $D \leq \frac{\rho(S^*, T^*)}{8(1+\varepsilon)\log_{1+\varepsilon} n}$ and*

$$\frac{\sqrt{|S^*|/|T^*|}}{(1+\varepsilon)} \leq z \leq \sqrt{|S^*|/|T^*|}.$$

*Proof.* Consider $(S_i, T_i)$ after running Algorithm 3. Then, in the best-case scenario, all the edges in its induced subgraph, $E_G(S_i, T_i)$, were used to determine the peeling of its vertices for later vertex sets. However, because the degree of a vertex $v$ is reset to 0 every time $v$ moves to a higher level, information about edges incident to $v$ up to that point is – informally speaking – erased; we call such edges *ignored*. At most $i \cdot k_S$ incident edges to a vertex in $S_i$ are ignored, and, similarly, at most $i \cdot k_T$ incident edges to a vertex in $T_i$ are ignored. Hence, throughout the entire algorithm, each vertex in $S_{2\log_{1+\varepsilon} n}$ ignores at most $2k_S \log_{1+\varepsilon} n$ edges incident to it and each vertex in $T_{2\log_{1+\varepsilon} n}$ ignores at most $2k_T \log_{1+\varepsilon} n$ edges incident to it.

Consider the induced subgraph on $(S^*, T^*)$. Since $(S^*, T^*)$ is the densest subgraph, we observe that all vertices in $S^*$ have degree at least $8k_S \log_{1+\varepsilon} n$ and all vertices in $T^*$ have degree at least $8k_T \log_{1+\varepsilon} n$ by following the proof of Lemma 3.1 and using the bounds on $D$ and $z$. We claim that some non-empty subsets of these vertex sets will remain in $(S_{2\log_{1+\varepsilon} n}, T_{2\log_{1+\varepsilon} n})$, which is enough to prove the lemma. Rather than letting edges be ignored throughout the algorithm, we assume they're all ignored *at the beginning*: $2k_S \log_{1+\varepsilon} n$ edges incident to each vertex in $S^*$ are ignored and $2k_T \log_{1+\varepsilon} n$ edges incident to each vertex in $T^*$ are ignored. Then, we look at how vertices within the induced subgraph on $(S^*, T^*)$ are peeled over time.

Notice that
$$|E_G(S^*, T^*)| \geq (8k_S \log_{1+\varepsilon} n)|S^*| = (8k_T \log_{1+\varepsilon} n)|T^*|$$
and so with the edges that are ignored, the number of remaining edges is lower bounded by

$$|E_G(S^*, T^*)| - (2k_S \log_{1+\varepsilon} n)|S^*| - (2k_T \log_{1+\varepsilon} n)|T^*|$$
$$\geq (4k_S \log_{1+\varepsilon} n)|S^*| = (4k_T \log_{1+\varepsilon} n)|T^*|.$$

When peeling is performed on this subgraph, all the vertices that are peeled each cause at most $k_S$ edges to be removed, if it is from $S^*$, and at most $k_T$ edges to be removed, if it is from $T^*$. If we assume that all the vertices are peeled, at most $2k_S|S^*| = 2k_T|T^*|$ edges will be removed, and the graph must be empty. However, this is smaller than the lower bound on the number of edges above, so it is impossible to peel all the vertices. Therefore, we end up with a non-empty subgraph, and the vertices in this subgraph will not be removed throughout the entire algorithm. $\square$

**Theorem 5.1.** *There exists a single-pass semi-streaming algorithm that attains an $O(\log n)$-approximation of the directed densest subgraph.*

*Proof.* Let $\mathcal{A}_{\text{STREAM}}$ be our semi-streaming algorithm. We now describe it and provide its analysis.

**Algorithm description.** $\mathcal{A}_{\text{STREAM}}$ runs Algorithm 3 on all $1 \leq D \leq n$ and $\frac{1}{\sqrt{n}} \leq z \leq \sqrt{n}$ in parallel using powers of $1 + \varepsilon$. Then, out of all the outputs of Algorithm 3 where the vertex sets are both non-empty, we pick the one corresponding to the largest $D$ as the final output of $\mathcal{A}_{\text{STREAM}}$.

**Algorithm memory.** Note that Algorithm 3 does not need to store $(S_i, T_i)$ for all $0 \leq i \leq 2\log_{1+\varepsilon} n$ but only needs to store consecutive sets to compare their sizes. Therefore, Algorithm 3 uses $O(n)$ memory through this implementation detail. Additionally, we have $O(\log^2 n)$ total guesses on $D$ and $z$. Each of these guesses is a copy of all the variables in Algorithm 3, resulting in $O(n \log^2 n)$ memory in total.

**Algorithm approximation.** Let the directed densest subgraph have vertex sets $(S^*, T^*)$. Note that because we only update $d(u)$ for an edge $(u, v)$ or $(v, u)$ with $l(u) < l(v)$, this ensures that a vertex that is not removed must have degree at least $D/(2z)$, if it is in $S$, and degree at least $Dz/2$, if it is in $T$. Therefore, if Algorithm 3 outputs non-empty vertex sets for $D \geq \frac{\rho(S^*, T^*)}{8(1+\varepsilon)^2 \log_{1+\varepsilon} n}$, we follow a similar argument as the proof of Algorithm 1 to see that these vertex sets would be an $O(\log n)$-approximation of the directed densest subgraph.

Specifically, there exists a guess on $D$ and $z$ such that

$$\frac{\rho(S^*, T^*)}{8(1+\varepsilon)^2 \log_{1+\varepsilon} n} \leq D \leq \frac{\rho(S^*, T^*)}{8(1+\varepsilon) \log_{1+\varepsilon} n} \text{ and } \frac{\sqrt{|S^*|/|T^*|}}{(1+\varepsilon)} \leq z \leq \sqrt{|S^*|/|T^*|}.$$

Using Lemma E.1, we know that all vertex sets $(S_i, T_i)$ in Algorithm 3 for this guess on $D$ and $z$ will be non-empty and so Algorithm 3 will output non-empty vertex sets. Since $\mathcal{A}_{\text{STREAM}}$ outputs the non-empty vertex sets corresponding to the largest $D$, they must satisfy $D \geq \frac{\rho(S^*, T^*)}{8(1+\varepsilon)^2 \log_{1+\varepsilon} n}$ and results in an $O(\log n)$-approximation. $\square$

# F  Additional streaming experiments

We run the algorithms on the general directed graphs. We plot the results of our algorithm run on various values of $\varepsilon$ ranging from 0.2 to 2 while we ran [BKV12] on $\varepsilon = 0.2$. Even though most of these datasets have edges sorted by their endpoints, which can be considered adversarial for our algorithm, we see from Figure 4 that their approximations are not far off from [BKV12]. Our maximum densities are within a factor of around 2 from [BKV12], which is much closer than our theoretical guarantees. We also consider a less adversarial order where we randomize the order of the edges. With a randomized order, we also see in Figure 4 that the computed densities increase significantly, almost matching those of [BKV12]. Figure 5 contains the update time plots for the remaining general datasets.

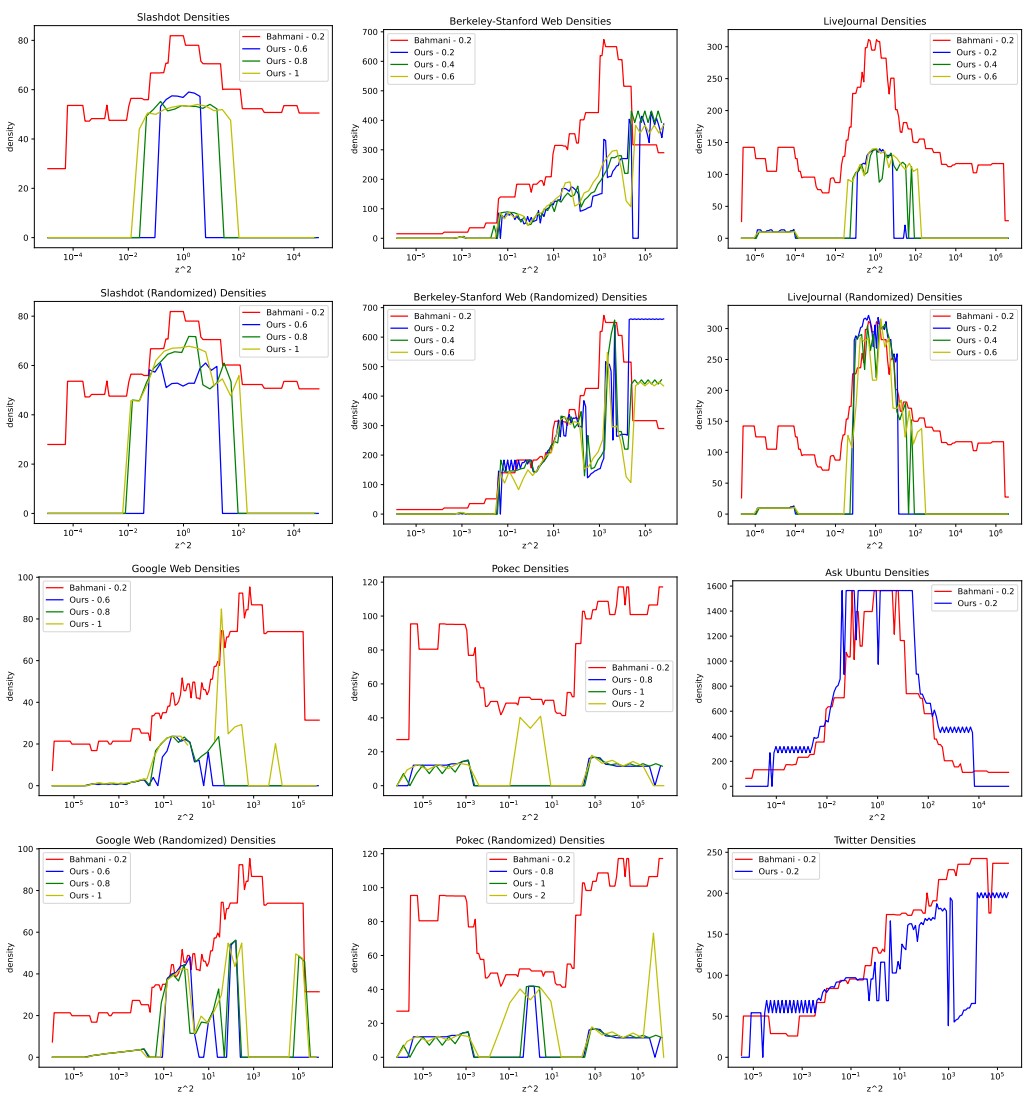

Figure 4: Density as a function of $z^2$ for general datasets and remaining temporal datasets.

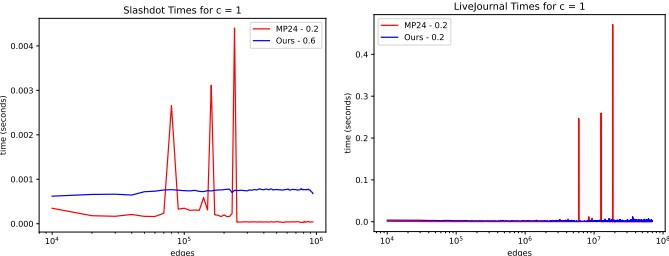

Figure 5: Update time between processing edges of the stream for remaining general datasets.

# G  MPC experiments

We compare our MPC algorithm to the MPC algorithm for directed graphs from [BKV12]. Specifically, we look at their density plots as well as how many sublinear MPC rounds the algorithms take. We use $\delta = 0.4, 0.6, 0.8$, where each machine has $n^\delta$ memory, and $\varepsilon = 0.6$, the approximation parameter for both algorithms. As we can see from Figure 6, our algorithm attains densities that

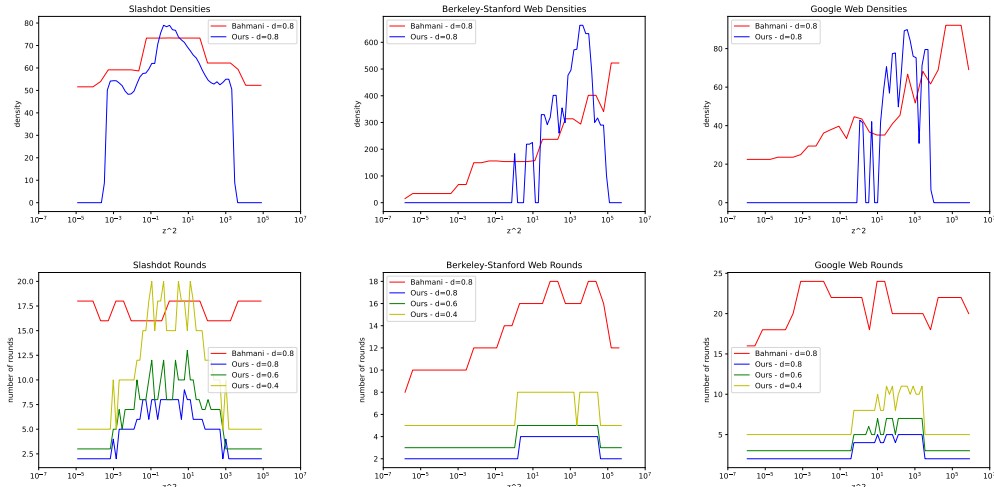

Figure 6: Density and number of MPC rounds as a function of $z^2$ for $\delta = 0.4, 0.6, 0.8$, $\varepsilon = 0.6$. The used datasets are described in Section 6.

match [BKV12]. Additionally, it does so using less than half the number of rounds when having the same amount of memory. It continues to use significantly less rounds for smaller amounts of memory and only matches the number of rounds for the Slashdot dataset where our algorithm uses a quadratically smaller amount of memory per machine compared to what is used by [BKV12]. This is a significant improvement in the number of rounds in practice and reflects our improvement in the algorithm's theoretical upper bound.

**Remark.** These experiments were not executed on actual large-scale frameworks, but simulated in a centralized setting. The rounds in plots correspond to the number of rounds taken in those MPC simulations.

