# OpenReview forum: "New Parallel and Streaming Algorithms for Directed Densest Subgraph"
_NeurIPS.cc/2025/Conference — NeurIPS 2025 poster_

### Official Review · Reviewer_hKXC · 2025-06-05

**Clarity:** 2
**Significance:** 3
**Originality:** 3
**Rating:** 4
**Confidence:** 4

**Summary:**

This paper studies the directed densest-subgraph (DS) problem in the Massively Parallel Computation (MPC) model and the streaming model. Let $n$ and $m$ be the numbers of vertices and edges of the input directed graph, respectively.

- In the MPC model, the authors present a $\tilde{O}(\sqrt{\log n})$-round algorithm that returns a $(2+\epsilon)$-approximation while using sub-linear (i.e., $O(n^{\delta})$ for some $\delta<1$) memory per machine. This result improves previous works which needs either $\Omega(\log n)$ rounds or $\tilde{\Omega}(n)$ memory, but at the cost of a looser approximation ratio.
- In the streaming model, this paper introduces the first single-pass deterministic streaming algorithm for directed DS on arbitrary streams. It uses $\tilde{O}(n)$ memory and achieves an $O(\log ⁡n)$-approximation.

**Questions:**

See "weaknesses"

**Ethical Concerns:**

["NO or VERY MINOR ethics concerns only"]

**Final Justification:**

This paper provides new algorithms with theoretical guarantees for the DS problem in the MPC setting and streaming setting. These results will be of interest to the theory and big-data processing communities. That said, I do not consider this paper a must-accept for an AI&ML-focused conference.

**Limitations:**

yes

**Paper Formatting Concerns:**

No major formatting issues

**Quality:**

3

**Strengths And Weaknesses:**

**Strengths**: This is a theoretical paper. These results are well-motivated, and strengthen existing bounds in meaningful ways. Moreover, the paper is clearly structured and well written.

**Weaknesses  and suggestions:** Please provide more specifics in the Introduction about the MPC algorithm, including the total number of machines used and the scheme used to partition the graph across them. In addition, I recommend adding two summary tables: one for existing MPC algorithms and another for streaming algorithms for tackling directed DS. Each table should list the key metrics for every algorithm (e.g., round complexity, memory per machine, and approximation ratio).

---

> ### Author Rebuttal · Authors · 2025-07-29
>
> Thank you for your feedback. For the introduction about the MPC algorithm, we will mention that the total number of machines is $n$, the number of vertices in the graph, and the edges are arbitrarily partitioned across the machines. As for your suggestion on the tables, we agree that it would make comparisons with previous works more clear and will add them in the final version of our paper.

---

> > ### Comment · Reviewer_hKXC · 2025-08-04
> >
> > Thanks for the response. I will keep my score.

---

### Official Review · Reviewer_jvHZ · 2025-06-27

**Clarity:** 3
**Significance:** 3
**Originality:** 3
**Rating:** 5
**Confidence:** 3

**Summary:**

This paper studies the directed densest subgraph (DS) problem under the MPC model and the semi-streaming model. In this problem, given a directed graph $(V, E)$, the goal is to find subsets S, T of V that maximize $\frac{|E(S, T)|}{\sqrt{|S| \cdot |T|}}$. The algorithms for both models are built on top of a simple "base" DS algorithm, which combines the standard iterated peeling procedure with a new stopping rule. In the MPC, the base algorithm is simulate via graph exponentiation. To apply the known simulation techniques, the authors design graph sparsification steps (e.g., freezing high-degree vertices and sampling edges) to control the neighborhood size of the graph. In semi-streaming, the authors make a key observation that the base algorithm can be simulated by maintaining a state vector recording how many peeling iterations each vertex survives. The MPC algorithm achieves $(2+\varepsilon)$-approximation in $\tilde{O}(\sqrt{\log n})$ rounds, using sublinear memory per machine. This matches the undirected bound and improves over the previous directed memory bound of $\tilde{\Theta}(n)$. The semi-streaming algorithm provides the first bound for arbitrary streams.

**Questions:**

1. It seems that the factor 2 in the (2+eps)-approximation MPC comes from the fact that the subgraph condition gives only 2-approximation.
- Is this relates to the fact that to handle directed graph, this paper ''duplicates'' the vertices to produce an undirected bipartite graph?
- Does the existing (1+eps)-approximation algorithm for *directed* graph use a similar subgraph condition?

2. In the context of semi-streaming, for undirected DS, [EHW15] provide a single-pass algorithm that guarantees a $(1+\varepsilon)$-approximation. The authors mention that their algorithm for directed DS can also be adapted to approximating the undirected DS. I assume this results in worse approximation factor compared to [EHW15], but do the authors observe any other advantage over them?

**Ethical Concerns:**

["NO or VERY MINOR ethics concerns only"]

**Final Justification:**

I didn't have major questions previously. My questions were mostly for curiosity, and the authors' answers are satisfying. So I will maintain my score.

**Limitations:**

Yes, the limitations are clearly discussed. The authors also point out multiple potential future directions.

**Quality:**

3

**Strengths And Weaknesses:**

**Strength**:

1. I like the general presentation of this paper: it starts with giving a simple ''base algorithm'' that is later used as a skeleton for both the MPC and the semi-streaming algorithm.
\
Although most proofs are delayed to the appendix due to the page limit, I think the high-level ideas of both algorithms and analysis are explained clearly in the main body of the paper. When describing the algorithms the authors carefully discuss the intuition behind each ingredient of the algorithm. And when outlining the analysis, they detailedly explain how lemmas interlock with each other and together lead to the main result.

2. While both algorithms are derived from the simple "base algorithm", lifting it to the MPC / semi-streaming settings involves ideas that feels nice and non-trivial.

- In the MPC case, to apply the graph exponentiation technique for simulation, the authors carefully control the size of neighborhoods by designing steps to freeze high-degree vertices and sample edges appropriately.

- In the semi-streaming case, I find the observation of reducing the simulation to computing a vertex-vector which records the peeling iteration for each vertex interesting. And I'm particularly surprised by how the algorithm contains a seemingly crude ''degree resetting'' procedure but still manage to achieve a log(n)-approximation.

**Weakness**:

- The main weakness that I can think of is both approximation factors achieved by this paper, $(2+\epsilon)$ and $O(\log n)$, can be a bit unsatisfying. However, as mentioned in the **Summary** above, these factors are already noteworthy improvements over states of the arts. So I find the contribution of this paper clear.

---

> ### Author Rebuttal · Authors · 2025-07-29
>
> Thank you for your comments. We address the questions provided below:
>
> 1. The factor of 2 in the approximation purely comes from the subgraph condition and is not related to the duplication of vertices to produce an undirected bipartite graph. The existing $(1+\epsilon)$-approximation algorithms do not use a similar subgraph condition. [EHW15] entirely uses one large uniform sample of the graph in order to approximate the directed DS. [BGM14] uses a technique called the multiplicative weights update method to approximately solve the directed DS LP. Both of these $(1+\epsilon)$-approximation algorithms use different techniques than our subgraph condition.
>
> 2. Yes, our approximation factor would be $O(\log n)$ compared to their $(1+\epsilon)$. However, their algorithm uses $\tilde{\Theta}(n^{1.5})$ memory, which is more than our $\tilde{O}(n)$ memory. So, their algorithm doesn’t fit within the memory constraints of the semi-streaming setting. Additionally, as mentioned in the paper, our algorithm is the first deterministic single-pass algorithm for approximating the undirected or directed DS as far as we know. These are the main advantages over [EHW15].

---

> > ### Comment · Reviewer_jvHZ · 2025-08-04
> >
> > Thank you for your response and answers to my questions! I will maintain my score.

---

### Official Review · Reviewer_o9zx · 2025-06-30

**Clarity:** 3
**Significance:** 3
**Originality:** 3
**Rating:** 5
**Confidence:** 3

**Summary:**

This paper presents new algorithms for finding approximate densest subgraphs in directed graphs within Massively Parallel Computation (MPC) and semi-streaming models. The proposed approaches achieve a $(2+\varepsilon)$-approximation in $\tilde{O}(\sqrt{\log n})$ MPC rounds with sublinear memory per machine and an $O(\log n)$-approximation in a single pass in the semi-streaming setting, significantly improving upon prior results. Empirical evaluations demonstrate that the algorithms are efficient in practice, with the semi-streaming method achieving better-than-theoretical guarantees on temporal datasets and the MPC algorithm requiring fewer rounds than existing methods.

**Questions:**

Q1. Please add more experiments regarding the MPC case.
Q2. Please add more explaination regarding the proposed algorithms to improve the readability.

**Ethical Concerns:**

["NO or VERY MINOR ethics concerns only"]

**Final Justification:**

The authors have addressed most of my concerns.

**Limitations:**

Yes

**Quality:**

3

**Strengths And Weaknesses:**

S1. The paper introduces novel algorithms for the directed densest subgraph problem that significantly improve upon existing methods, particularly in the Massively Parallel Computation and semi-streaming models, offering better approximation guarantees and efficiency.
S2. It provides strong theoretical results, including a $(2+\varepsilon)$-approximation in $\tilde{O}(\sqrt{\log n})$  MPC rounds and an $O(\log n)$-approximation in a single pass for semi-streaming, which bridge the gap between directed and undirected densest subgraph problems.
S3. The empirical evaluations show that the proposed algorithms perform well in practice, often exceeding theoretical expectations, especially in the semi-streaming context.

W1. Some steps in Algorithm 2 involve complex sampling and degree-thresholding mechanisms, and the explanations—although sufficiently technical—could benefit from more intuitive discussion, especially regarding relation to graph sparsification and the effect of 'freezing' high-degree vertices.
W2. Most experimental comparisons are offered for the semi-streaming algorithm. While this supports the main claims, the MPC algorithm’s empirical performance is only briefly mentioned and detailed results are relegated to the appendix, limiting the main-paper evidence for the MPC improvements.

---

> ### Author Rebuttal · Authors · 2025-07-29
>
> Thank you for the helpful suggestions. We address the suggestions provided below:
>
> Q1: We have executed additional experiments for MPC. Our additional experiments show that using less memory per machine slightly increases the number of rounds for our algorithm -- as expected -- but it is still significantly less than the number of rounds used by [BKV12]. The conference rules suggest we do not send a link to those plots, but we will include the plots in our final version.
>
> Q2: We do realize that Algorithm 2 involves technical tools, and we will add more intuitive explanations of these tools in the final version of the paper. In particular, we will expand more on the idea of “freezing” high-degree vertices and how that goes hand in hand with our sampling scheme.

---

### Official Review · Reviewer_bSSJ · 2025-07-02

**Clarity:** 3
**Significance:** 3
**Originality:** 3
**Rating:** 5
**Confidence:** 3

**Summary:**

This paper proposes new algorithms for approximating directed densest subgraphs in Massively Parallel Computation (MPC) and semi-streaming models. Specifically, in the MPC model, the paper provides a $(2+\epsilon)$-approximation in $\tilde{O}(\sqrt{\log n})$ rounds in the sublinear memory regime, which improves the best-known results and bridges the gap between the directed and undirected cases. In the semi-streaming model, the paper provides a (deterministic) $O(\log n)$-approximation using a single pass, which is the first (deterministic) single-pass semi-streaming algorithm for this problem. Both algorithms are based on an elegant peeling-based algorithm, but its efficient implementation in MPC and streaming models are non-trivial, and involves several novel insights developed in this paper. Last but not least, the paper empirically evaluates the effectiveness of the proposed algorithms on several large real-world datasets.

**Questions:**

Q1: Algorithm 3 resets the vertex degree to 0 when it moves to a higher level, assuming the worst case. Could this be improved to achieve a better approximation under the assumption that the stream arrives in random order?

Q2: Can the semi-streaming algorithm (i.e. Algorithm 3) be extended to work in dynamic streams, where both edge insertions and deletions are allowed?

**Ethical Concerns:**

["NO or VERY MINOR ethics concerns only"]

**Final Justification:**

NA

**Limitations:**

Yes

**Quality:**

3

**Strengths And Weaknesses:**

Strengths:

1: Computing densest subgraphs in directed graphs is a fundamental problem and would be of interest to the NeurIPS community. The paper is clearly structured and very well written.

2: The paper provides new algorithms in the MPC and semi-streaming models, improving upon the best-known results. The analyses are non-trivial, technically sound, and involve several novel insights.

3: The paper conducts empirical evaluations on several large real-world networks, demonstrating the effectiveness and efficiency of the proposed algorithms.

Weaknesses:

1: The proposed semi-streaming algorithm appears to work only in insertion-only streams. It would strengthen the work to extend it to dynamic streams, which represent a more general setting.

2: More comprehensive evaluations could be conducted in the experiments, e.g. the space usage of the algorithms.

3: Some minor issues:

-- Line 63: show -> shows

-- Line 64: require -> requires

-- In addition, it would be helpful to include a table summarizing existing results and the new results proposed in this paper for clearer comparison.

---

> ### Author Rebuttal · Authors · 2025-07-29
>
> Thank you for the helpful suggestions. After receiving your feedback, we have executed additional experiments for MPC. Our additional experiments show that using less memory per machine slightly increases the number of rounds for our algorithm -- as expected -- but it is still significantly less than the number of rounds used by [BKV12]. The conference rules suggest we do not send a link to those plots, but we will include the plots in our revised version. Additionally, we will add a table of previous results to our final paper in order to allow for clearer comparisons. We also address the questions provided below:
>
> Q1: If the stream arrives in a random order, we would be able to leverage how sets of edges in the stream are uniform samples of the entire graph. MP24 already provides a result for this setting, though, attaining a $(2+\epsilon)$-approximation using this random order stream property.
>
> Q2: Thank you for this excellent question! That is something we considered, but the ways of extending our algorithm to allow edge deletions makes it difficult to prove that the $O(\log n)$-approximation still holds. For example, edge deletions should be able to decrease the level of a vertex, but then it is unclear how to set its degree within this lower level. Handling this by resetting the degree back to 0 would result in losing too much information about vertex’s neighbors. So, a different idea is necessary. This is one of the many complications that arise if we still want to maintain the simplicity of our semi-streaming algorithm. Nevertheless, we find this to be a fantastic direction for future work!

---

### Official Review · Reviewer_nQ4e · 2025-07-05

**Clarity:** 2
**Significance:** 3
**Originality:** 2
**Rating:** 3
**Confidence:** 4

**Summary:**

This paper studies the problem of approximating the densest subgraph in directed graphs within the MPC and streaming models. There are three main categories of known results for the densest subgraph problem (both directed and undirected) in these computational settings.

First, for undirected graphs, Mitrovic and Pan \cite{MP24} proposed an $O(\sqrt{\log n})$-round MPC algorithm that requires $\tilde{O}(n)$ space per machine, which is linear in the number of nodes.

Second, in the streaming model, Esfandiari, Hajiaghayi, and Woodruff \cite{EHW15} (SPAA 2015) developed streaming algorithms that handle both insertions and deletions of edges. Their method samples $\tilde{O}(n)$ edges uniformly at random and then applies an offline densest subgraph algorithm to the sampled set at the end of the stream.

For directed graphs, Mitrovic and Pan \cite{MP24} also presented an MPC algorithm with $O(\log n)$ rounds. The streaming algorithm of \cite{EHW15} can be generalized to the directed case, but this requires $O(n^{1.5})$ space, which is no longer near-linear. Alternatively, the method of \cite{MP24} can produce a streaming algorithm with near-linear space complexity, but only if the input edges arrive in random order (i.e., the stream is randomly permuted).

In contrast, this paper presents new MPC and streaming algorithms for the directed densest subgraph problem that outperform the prior results of \cite{EHW15} and \cite{MP24}. In the MPC model, the authors propose an $O(\sqrt{\log n})$-round algorithm. However, throughout the paper (including Result 1 and Theorem 4.2), the authors repeatedly state that their algorithm uses “sublinear” space per machine without specifying the exact bound. This makes it hard to properly evaluate the result, and the exact space complexity should be clearly stated.

The second main contribution, described in Result 2 and Theorem 5.1, is a single-pass deterministic semi-streaming algorithm achieving an $O(\log n)$-approximation for the directed densest subgraph. Although the approximation factor is not constant, this result is interesting because it appears to be the first deterministic streaming algorithm for this problem.

Here are two additional suggestions to improve the paper:

- The authors should cite and discuss the work of Monika Henzinger (STOC 2015), which is a pioneering contribution in this area and should be acknowledged.

- The paper introduces a non-standard term for the model where edges arrive in random order; the common name for this setting is random order streams. Using the standard terminology would make the presentation clearer.

**Summary**

Overall, the paper presents interesting and potentially impactful results for the directed densest subgraph problem in MPC and streaming models. However, the presentation could be significantly improved by clearly stating the exact space complexity of the algorithms, structuring the related work discussion more carefully, and aligning terminology with standard usage in the literature. Addressing these points would make the contribution clearer and strengthen the paper.

**Questions:**

**Questions and Suggestions**

- Could the authors clearly specify the exact space complexity per machine used by the MPC algorithm? Using only the term “sublinear” is too vague; please state the precise asymptotic bound (e.g., $\tilde{O}(n^\alpha)$ for some explicit $\alpha < 1$).

- Can the authors elaborate on the technical novelty compared to prior work, especially \cite{MP24} and \cite{EHW15}? What new ideas or analysis allow achieving the improved number of MPC rounds or the deterministic streaming result?

- Please discuss the pioneering work by Monika Henzinger (STOC 2015), and explain how this paper relates to or builds on that contribution.

**Ethical Concerns:**

["NO or VERY MINOR ethics concerns only"]

**Limitations:**

Yes

**Quality:**

2

**Strengths And Weaknesses:**

**Strengths**
- The paper studies the directed densest subgraph problem, which is an important and timely topic in streaming and MPC models.
- Proposes an $O(\sqrt{\log n})$-round MPC algorithm, improving over the previous $O(\log n)$-round result by \cite{MP24}.
- Presents what appears to be the first deterministic semi-streaming algorithm for approximating the densest subgraph in directed graphs, which could be of independent theoretical interest.

**Weaknesses**

- The paper does not clearly specify the exact space per machine used in the MPC algorithm, repeatedly describing it only as “sublinear,” which makes the result hard to evaluate and compare.

- The discussion of related work is limited and does not include important prior contributions (e.g., Henzinger STOC 2015).

- The paper uses non-standard terminology for “random order streams,” which could confuse readers.

---

> ### Author Rebuttal · Authors · 2025-07-29
>
> Thank you for the feedback and detailed reading of our paper. We address the questions/suggestions provided below:
>
> - The value of $\delta$ provided to Algorithm 2 specifies that each machine will use $O(n^\delta)$ memory, which is then shown in the proof of Theorem 4.2. We will make this more clear by adding it to our result and theorem statements; thanks for pointing that out.
>
> - The main technical comparison, which is discussed in Section 3.1, is that our algorithm uses new stopping rules compared to other peeling algorithms (BKV12, GLM19, MP24). These rules give us an efficient, simple peeling algorithm, which enables us to improve existing results for MPC and semi-streaming. As for EHW15, their work is limited to a single uniform sample of the entire graph, while our MPC algorithm takes many samples in order to approximate degrees for our base peeling algorithm. This is not exactly novel but combined with our new base algorithm, it gives us the new results presented in our paper.
>
> - From our understanding, Bhattacharya et al. (STOC 2015) is a dynamic algorithm that outputs an $(8+\epsilon)$-approximation of the density value and does not provide vertex sets for the approximate directed DS. Additionally, it has an amortized update time of $\tilde{O}(1)$, the worst-case update time that is polynomial in $n$, and uses linear memory $O(m)$, so it does not fit within the memory of the semi-streaming model. Sawlani & Wang (STOC 2020), which we cite in our submission, provides a dynamic algorithm that outputs a $(1+\epsilon)$-approximation including the vertex sets of the approximate directed DS. Nevertheless, we agree that we should cite and discuss Bhattacharya et al. (STOC 2015); we will do so in our final version by including a refined version of the following paragraph.
> - As for the techniques, the $(\alpha, \delta, L)$-decomposition introduced by Bhattacharya et al. (STOC 2015) is an extension of a fixed d-core which is a classical idea for approximating the DS. Our paper revolves around calculating something similar, as reflected in Lemma 3.2, but the way we compute it using peeling is novel because of the stopping conditions described in Section 3.1; in short, it allows us to keep a fixed threshold of peeling while guaranteeing an approximation in a small number of iterations. Bhattacharya et al. (STOC 2015) does not maintain this fixed threshold and actually allows relaxations in the threshold, specified by the $\alpha$ parameter, in order to keep the number of iterations small. Additionally, our algorithm outputs an approximate DS by comparing vertex set sizes, while Bhattacharya et al. (STOC 2015) only proves an existence of some vertex sets that attain their approximation (their result presented in Corollary 6.7). These reflect some of the major differences between the algorithms that allow us to attain the new results in our paper.

---

> > ### Comment · Reviewer_nQ4e · 2025-08-07
> >
> > Dear Authors,
> >
> > Thank you for your responses. I have a few remaining questions.
> >
> > First, why didn't you include the space per machine, the total space, and the total work time in the statements of your theorems? When developing MPC algorithms, clearly stating all these complexities is very important for understanding the practical implications and efficiency of your approach.
> >
> > Additionally, could you clarify the space usage of the streaming algorithm presented in [MP24] for random order streams? Also, what would be the approximation factor and space usage of your algorithm for random order streams?
> >
> > I look forward to your clarifications.

---

> ### Author Response · Authors · 2025-08-07
>
> Thank you for the additional comments.
>
> > Space per machine:
>
> We agree that this could be made more explicit. Our theorems and preliminaries state that we work in the sublinear MPC regime, which by standard definition means that each machine has $O(n^\delta)$ space for an arbitrarily small constant $\delta > 0$. In this model, $\delta$ is a system parameter rather than something fixed by the algorithm, which is why we did not commit to a specific value. That said, we understand that not all readers may be familiar with this convention, and we will revise the theorem statement to explicitly include that each machine has $O(n^\delta)$ memory for some constant $\delta > 0$.
>
> > Why didn't you include ... total space and total work time in theorems?
>
> There is no specific reason—this was purely an omission on our side. As a direct consequence of the design, the total number of machines used is $O(n + m / n^\delta)$, so the total space is $O(n^{1+\delta} + m)$. We will include this in the theorem statement.
>
> As for the total work/time, each round of our algorithm performs near-linear work. In the literature on sublinear MPC algorithms, round complexity is the main bottleneck and is typically the focus; total time per round is rarely spelled out unless it is superlinear. Nonetheless, to avoid ambiguity, we will include a line in the preliminaries explicitly stating:
> "Even though the running time in definition of the MPC model is allowed to be arbitrarily large, the running time of our MPC algorithm is near-linear per round."
>
> > Additionally, could you clarify the space usage of the streaming algorithm presented in [MP24] for random order streams? Also, what would be the approximation factor and space usage of your algorithm for random order streams?
>
> The space usage in [MP24] is $O(n \log^2 n)$. Our semi-streaming algorithm has the same space complexity and does not change when applied to random-order streams; however, unlike [MP24], it is designed to work even under adversarial edge arrival. As a result, when run on a random-order stream, our algorithm still achieves an $O(\log n)$-approximation while using $O(n \log^2 n)$ memory.
>
> Beyond random vs adversarial order, an important distinction lies in the worst-case update time. The algorithm in [MP24] relies on global peeling phases, which can lead to high and unpredictable update times. In contrast, our algorithm guarantees worst-case update time of $O(\log^2 n)$ per edge. This difference is evident in practice, as shown in Figure 3 of our submission.

---

### Note · Authors · 2025-08-11

We thank all the reviewers once again for their feedback. Based on this feedback, we plan to make the following changes to our paper:
- Include more MPC experiments with varying memory per machine
- Add tables that summarize previous results compared to our work
- Revise theorem statements to be more explicit about the sublinear MPC model
- Include citation of Bhattacharya et al. (STOC 2015)

Additionally, to conclude our discussion with Reviewer nQ4e, we emphasize that Bhattacharya et al. (STOC 2015) does not subsume our approach. Our approach is mainly inspired by Bahmani et al. (VLDB 2012), which is cited mutliple times in our paper, and we're able to achieve the results presented in our paper through our novel stopping rules; we refer the reviewer to Section 3.1 of our paper and also to other reviews, e.g. Reviewer jvHZ. These stopping rules are the main technical contribution that allows our algorithms to improve the state of the art.

---

### Decision · Program_Chairs · 2025-09-17

**Decision:**

Accept (poster)

**Comment:**

This paper studies the densest subgraph problem for directed graphs in various big data models, achieving a $(2+\varepsilon)$-approximation in $\tilde{O}(\sqrt{\log n})$ MPC rounds and a $O(\log n)$-approximation in the semi-streaming model, i.e., using $\tilde{O}(n)$ space. By comparison, a previous work achieves $O(n^{1/6})$-approximation in the semi-streaming model. Although there were concerns about both the scale and the presentation of the experiments, all reviewers agreed that the theoretical results are interesting for both MPC and the streaming model.

I believe the reviewer feedback has resulted in a number of actionable items that would improve the quality of the paper and thus I would encourage the authors to incorporate these comments to strengthen the potential impact of the work.